# Private capital holding, financial awareness of government and steadying operation of banks: Evidence from China

Kun Xu[1], Jie Liu[2]*, Jiarong Li[3]

1 Institute of Chinese Financial Studies, Southwestern University of Finance and Economics, Chengdu, Sichuan, China, 2 School of Finance, Southwestern University of Finance and Economics, Chengdu, Sichuan, China, 3 School of Economics, Southwestern University of Finance and Economics, Chengdu, Sichuan, China

* liuj@smail.swufe.edu.cn

Data Availability Statement: All relevant data are within the paper and its Supporting information files.

Funding: Our research was funded by Humanities and Social Sciences Foundation of the Ministry of

## Abstract

Based on the panel data of 123 city commercial banks in China from 2007 to 2017, we use the dynamic panel system GMM estimation method to empirically test the impact of private capital holdings on the stability of city commercial banks. The results show that private capital holding improves the operating performance of city commercial banks, reduces the volatility of return on total assets, and is conducive to the stability of city commercial banks. Furthermore, the lack of financial awareness of local governments has led to the negative impact of private capital, that is, the stability of the banks has declined.

## Introduction

Since the reform and opening up for more than 40 years, Chinese private economy has undergone significant changes. The development of private enterprises has developed from weak to strong and from entity operation to capital operation. Some private capital begin to hold controlling shares of city commercial banks (CCBs), particularly, through holding controlling shares of shareholding bank, owing to the relationship of investment and financing between enterprises and banks and the huge monopoly profits of banking industry. Private capital became an important shareholder of CCBs along with private equity larger than state-owned equity by 2017. Generally, the profitability, robustness, rigid system and management structure of the CCBs are improved by holding controlling shares of shareholding bank from private enterprises which have strong competitiveness and high efficiency of corporate governance. However, private capital holding controlling shares of city commercial banks (PCSCCB) is leading to new issues, such as impact on the financial stability of the CCBs or be displaced by private capital. Obviously, it goes against the original intention of introducing private capital into the CCBs. Facing the new phenomenon of PCSCCB and its positive and negative effects on CCBs, it is necessary to do systematic research and to explore the underlying reasons in academic circles.

Scholars have been deep analyzing issues of private capital being a shareholder of banks and bank equity (nature and concentration) and steady operation. A number of literature focused

Education of China (No. 19YJC790162). The funder had no role in study design, data collection and analysis, decision to publish, or preparation of the manuscript.

**Competing interests:** The authors have declared that no competing interests exist.

on the motivation of private enterprise held banks shares [1, 2] and the impact of state-owned equity on bank performance and risk-taking [3, 4]. Recently, there are also literatures focusing on the impact of private shareholders on bank lending behavior [5, 6]. In fact, the PCSCCBs meet the financing needs of private enterprises while have an important impact on the operation of holding banks. However, current theoretical and empirical studies about the impact of private equity on bank performance and risk-taking are insufficient and lack a unified index.

We attempt to illustrate that the introduction of private capital is conducive to the release of the potential efficiency of the CCBs, but the result is closely linked with the financial awareness of the government. The stronger the financial awareness of government is, the stronger of supervision illegal operation on private capital holding CCBs will be, instead of decreasing financial efficiency as some arguments believed [7].

In order to empirically analyze the external impact mechanism of private capital holding on the steady operation of the CCBs, we set up the index of government financial awareness based on the data of the top ten shareholders of 123 CCBs from 2007 to 2017 in China.

Our contribution involves in taking the bank operation and risk bearing into the index of bank robustness, setting up the index of government financial awareness for exploring the external mechanism, and decomposing the index of bank robustness for researching the influence of private capital holding and government financial awareness on the stability of the CCBs based on the essential characteristics of private controlling shareholders in the CCBs in the first time. This provides a research paradigm for the study of the sound operation of banks. More importantly, we have enriched the theory of bank corporate governance. We supplemented the relevant research on the ownership structure of banks from the perspective of private shareholders. In addition, we also demonstrated the role of the government in the banking system.

This study consists of seven parts. The first part includes the introduction, the second part provides the literature review and research hypothesis, the third part contains the research design, the fourth part consists of the empirical results and discussion, the fifth part provides the robustness test, and the sixth part is the endogeneity explanation, the last part outlines the conclusion and enlightenment.

## Literature review and research hypothesis

### Ownership structure and steady operation of banks

Ownership structure has an important influence on both bank governance mechanism and performance [8]. The effect of ownership structure linked with control and ownership nature on the value of banks has always been a controversial issue. Some believe the "interest coordination" hypothesis, that is, effectively supervising managers, improving the quality of loans and the operation of banks by equity concentration [9, 10]. Others propose the "tunneling" hypothesis which means that the major shareholders would erode the interests of other shareholders for pursuing the control right, and thus increase the bad loan and lower the operating performance because of the concentration of equity [11]. Another literature focuses on the role and influence of government holding on CCBs. It is a common phenomenon of government holding banks as the majority of controlling shareholders of CCBs are local governments [3]. The opinion of mainstream worry about decreasing bank operation efficiency and worsening bank risk bearing because government holding banks tend to provide loans to local state-owned enterprises and government financing platforms [4, 12, 13].

Most studies support the "hollow out" hypothesis of government holding banks. Compared with other countries, Chinese fiscal and financial institutional arrangements are special, which makes it difficult to restrict the government's "tunneling" behavior and the hidden cost of

government tunneling local banks is low. Conversely, private equity can significantly improve performance of banks. As a special form of private equity, private capital holding should pay the role in improving the performance and robustness of the CCBs. Before the subprime crisis, ownership structure (nature and concentration) was closely related to bank risk-taking [9]. After the subprime mortgage crisis, the research on ownership structure and bank risk-taking has made new progress. The nature of equity is no longer a factor that affects bank risk-taking. The deposit insurance system and other safety nets make banks with concentrated equity easier to bear individual and systemic risks [14, 15]. Therefore, private capital replace the local government as the controlling shareholder of the CCBs, which would improve the risk-taking of the CCBs. Based on these analyses, we propose research **Hypothesis 1**:

**Hypothesis 1**. Private capital holding is generally conducive to the stable operation of the CCBs through improving the operating performance and reducing the volatility of operation.

## Financial awareness of government, private capital holding, and steady operation of banks

Local protection and competition are the key to understanding Chinese economic growth. The financial awareness of government is shown by government's holding large-scale financial assets that is closely connected with the performance and risk management of banks.

At the beginning of the reform, in order to cooperate with the reform in the economic field, the financial field broke the "unified" banking system and the local finance began to develop. The local financial institutions represented by the city credit cooperatives were set up and developed rapidly under the promotion of the government. Local financial resources and the control right of local banks were dominated by local governments. To a large extent, local governments have obtained the right to allocate local financial resources which are regarded as a substitute of fiscal resources for a long time.

The trend of controlling over banking financial resources with government was accelerated by competing for winning promotion of government officers [7]. Local governments usually seek the holding CCBs to support the development of local state-owned enterprises, even if their inefficiency, because of the high return of control rights to local state-owned enterprises. However, with financing to poor efficient state-owned enterprises, local government intervention caused a large number of bad loans, and eventually "emptied" local banks or led to the accumulation of operation risk of the CCBs. The loan behavior of banks was seriously interfered by a large number of loans being invested in state-owned enterprises associated with the government.

Scholars discussed that excessive government intervention in bank loan would lead to the decline of bank loan quality and to seriously affect the stable operation of banks [16]. For the sake of limiting the inefficient expansion of local state-owned enterprises and prevent the accumulation of systemic financial risks from threatening economic security, the central government required local governments to accelerate the local financial reform and to encourage introducing competition mechanism in the financial market for the purpose of taking the local financial order back to normal and decreasing financial risks. Thus, the proportion of financial institutions held by local governments declined or withdraw and the financial awareness of local governments began to fade. At this time, the introduction of private capital would form a benign game of mutual restriction between local governments and private capital, which could improve the corporate governance level of city commercial banks (CCBs). Private capital holding could avoid the absolute control of state-owned shareholders over CCBs, and improved

the business performance of CCBs. Meanwhile, local governments still had the right to allocate important financial resources and restricted the operation of private capital holding CCBs, so that avoided the behavior of private capital "hollowing out" CCBs.

Yet, potential issues would be arisen, that is, the lower the local government's financial awareness, the greater the competition pressure in the financial markets. Because the aim is different between local governments and private controlling shareholder of CCBs. The board of directors and managers of CCBs controlled by the local government, for example, when facing the dual goals of profit assessment and political demands, choose more political demands as the primary aim and avoid the banks from engaging in high-risk projects for the purpose of operate steadily. Instead, the political demands of the board of directors and managers of private capital holding CCBs are weak and pursue profits as the main target, which is easy to produce a connection and actions with common interests, especially, it easier to engage in high-risk business activities drove by financial market competition and lead to poor or instable business performance of CCBs. In addition, the high-risk behavior of the private controlling shareholders will be exacerbated by the low financial awareness of local government makes the long-term due to lacking the regulatory and restrictive mechanism for the private controlling shareholders of CCBs. Anginer et al. [15] found that corporate governance with consistent actions is not favor of the stable operation of banks. Based on these analyses, we propose research **Hypothesis 2**:

**Hypothesis 2**. The government's financial awareness can moderate the effect of private capital holding. Specifically, the lower the government's financial awareness is, the more private capital holdings are detrimental to the profitability and the stability of city commercial banks.

## Data and methods

### Data sources

As of January 2023, there are a total of 125 city commercial banks in China (very few of them have come from the restructuring of our sample banks). The information disclosure of some banks is not perfect (their scale is also small), so we can't get their relevant annual reports, and finally we got 123 CCBs. Therefore, our sample represents almost all city commercial banks in China. CCBs began to disclose annual reports generally and in a standard manner in 2007, and some new city commercial banks have been established by restructuring several city commercial banks since 2018. So, we selected the annual panel data of CCBs from 2007 to 2017. The financial data and ownership structure of CCBs are derived from the annual reports of their official websites. The ownership and total asset scale data of insurance institutions, securities, funds, trusts and banks related to the construction of government financial awareness indicators are derived from the annual reports of these financial institutions. The data of GDP comes from the national bureau of statistics of China.

### Model specification

The current operation of the bank is continuous in time dimension. Which is inertia of the dependent variable [17, 18]. In addition, among the factors affecting the stability of banks, many unobservable bank characteristics are also related to dependent variable. To eliminate the impact of these factors, we have adopted the following dynamic panel regression model:

$$
\begin{aligned}
Ln\_Z_{i,t} &= \beta_0 + \rho_1 Ln\_Z_{i,t-1} + \beta_1 Priv_{i,t} + \beta_2 Finawa_{i,t} + \beta_3 Priv_{i,t} \times Finawa_{i,t} \\
&\quad + \theta X + year_t + \omega_i + \epsilon_{i,t}
\end{aligned}
\tag{1}
$$

**Table 1. Variable definitions.**

| Variable | Definition |
|---|---|
| Ln_Z | Stability of bank. We take Z-score to measure the stability of commercial banks draw on the experience of Michalak and Uhde [19]. $Z=(AROA+ACAR)/\sigma(ROA)$. Here, $AROA$ and $ACAR$ are the average values of $ROA$ and $CAR$ in three years (the previous year, the current year and the next year), and $\sigma(ROA)$ is the standard deviation of $ROA$ in three years. After the Z-score is obtained, variables are measured by the natural logarithm ($Ln\_Z$). The larger the $Ln\_Z$, the more stable the bank. |
| Priv | We refer to the research of Iannotta et al. [9] and Dong et al. [20] to set dummy variable, $Priv$, if the largest shareholder of city commercial banks is private capital, the value is 1, otherwise it is 0. |
| spriv | The sum of private ownership shares among the top ten shareholders. |
| Privl | A dummy, if the sum of private ownership shares among the top ten shareholders is greater than the sum of state-owned ownership shares, the value is 1; otherwise, it is 0. |
| Finawa | The financial awareness of local governments, $Finawa$, is a provincial indicator. In China, major financial institutions include banks, insurance institutions, securities, funds and trusts. According to the annual reports of various financial institutions in various provinces, we collect the information of ownership structure of each institution to obtain the amount of shares held by the local government, and calculate the assets held by the local government. Of course, we also need to add up the total assets of all financial institutions in the province to calculate the average share of financial assets held by local governments, namely, $Finawa$. It is calculated by the following formula:$Finawa_{i,t}=1$-Financial assets held by local governments$_{i,t}$ / Total local financial assets$_{i,t}$. The larger the $Finawa$, the less financial resources controlled by the local government, indicating that the financial awareness of the local government is weak. |
| Loan | The ratio of loan size to total assets. |
| NSCR | Henderson and Pearson [21] showed that shadow banking business has a significant impact on banking operations. Therefore, we selected the ratio of non-standard bond business to total assets, $NSCR$, as the proxy variable of shadow banking business of city commercial banks. |
| NPL | Non-performing loan ratio. |
| CAR | The capital adequacy ratio. |
| GDP | The logarithm of per capita GDP in the province where the bank is located. |

We use the systematic GMM estimation method for regression analysis to solve the endogeneity between the lag-term and the dependent variable.

## Variable description

The variables in Eq (1) are defined in Table 1. The dependent variable, $Ln\_Z_{i,t}$, is stability of bank $i$ in year $t$. The Z-scores is composed of the return on total assets ($ROA$), capital adequacy ratio ($CAR$) and volatility of $ROA$ ($\sigma(ROA)$). The independent variable, $Priv_{i,t}$, is a dummy variable, if the largest shareholder of city commercial banks is private capital, the value is 1, otherwise it is 0. $Finawa_{i,t}$ is the financial awareness of the local government where bank $i$ is located in year $t$. The purpose of the interaction item, $Priv_{i,t} \times Finawa_{i,t}$ is to test whether the government's financial awareness has a moderating effect on the impact of private capital on bank stability. $\beta_0$ is a constant term, $\omega_i$ is the banks' individual effect, $\epsilon_{i,t}$ is the error term. Adding control variables could solve the endogeneity problem caused by missing variables. $X$ are some control variables, including bank characteristics and macroeconomic characteristics. Characteristic variables at the bank level: $Loan$, $NSCR$, $NPL$, and $CAR$. Characteristic variables of regional macroeconomics: $GDP$. Because almost all CCBs have branches in many cities in their provinces, the GDP here is the provincial level. Of course, $year_t$ is year effect.

## Results and discussion

### Descriptive statistics

Table 2 reports descriptive statistics. The mean of $Ln\_Z$ is 1.944, the standard deviation is 0.675. This shows that the stability difference between banks is large. The mean of $Priv$ is 0.176. That is, 17.6% of the banks have private enterprises as their largest shareholders.

**Table 2. Descriptive statistics.** This table reports the descriptive statistics for the main variables. The sample period is from 2007 to 2017.

| Variables | Mean | SD | Min | Max | N |
|---|---|---|---|---|---|
| Ln_Z | 1.944 | 0.675 | 0.208 | 4.480 | 530 |
| Priv | 0.176 | 0.381 | 0.000 | 1.000 | 708 |
| Finawa | 0.528 | 0.220 | 0.000 | 1.000 | 1353 |
| Loan | 0.432 | 0.106 | 0.126 | 0.739 | 1137 |
| NSCR | 0.174 | 0.121 | 0.000 | 0.600 | 829 |
| NPL | 0.014 | 0.013 | 0.000 | 0.244 | 1141 |
| CAR | 0.135 | 0.042 | 0.034 | 0.596 | 1157 |
| GDP | 3.636 | 0.492 | 2.064 | 4.859 | 1353 |

By showing the change of *Ln_Z* with years in Fig 1, we can find that LnZ generally shows a trend of increasing in the early stage and decreasing in the later stage. However, when *Priv*=1, *Ln_Z* is higher. In Fig 2, the change trend of *ROA* is generally similar to that of *Ln_Z*, which increases in the early stage and decreases in the later stage. Coincidentally, when *Priv* = 1, the *ROA* is higher. The change trend of *CAR* also presents a similar situation as shown in Fig 3. But the difference is that when *Priv* = 1, *CAR* is not necessarily higher. Through visual observation of these graphs, it can be preliminarily concluded that the change of bank stability may be more related to the profitability.

## Stability test of variables

We judge whether the main variables are a stationary process according to the Fisher-ADF test proposed by Choi [22]. Table 3 reports this test result. It shows that the statistics (Inverse Chi-squared P) strongly reject the original hypothesis of panel unit root, so the main variables are a stationary process.

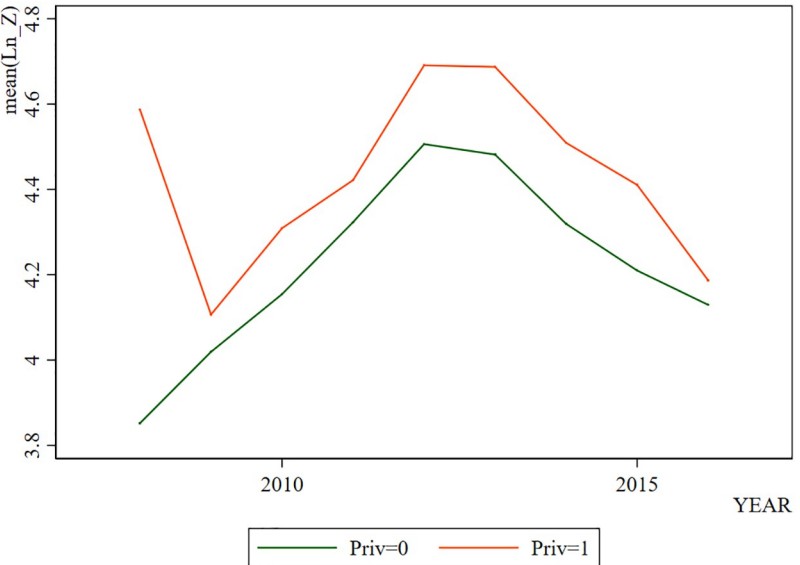

**Fig 1. *Ln_Z* in different years.**

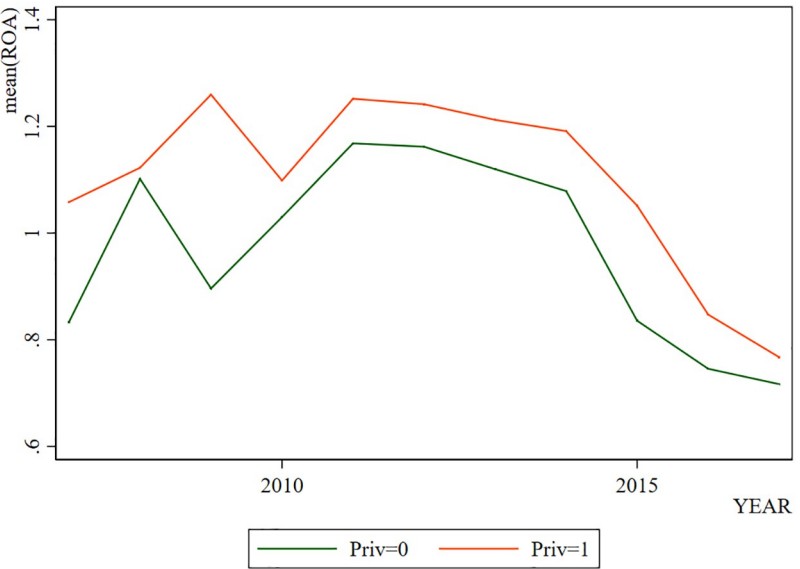

**Fig 2.** *ROA* **in different years.**

## Results

Based on Eq (1), we run a dynamic panel system GMM regression based on stability of bank, private capital holding and the moderating effect of government financial awareness. In fact, the Z-scores is composed of the return on total assets (*ROA*), capital adequacy ratio (*CAR*) and volatility of *ROA* ($\sigma(ROA)$). A higher *ROA* and *CAR* or a lower $\sigma(ROA)$ can increase the Z-scores. In order to clarify whether the change of Z-scores is caused by the change of return on total assets or capital adequacy ratio, we further regress the relationship between private holding and each component (*ROA*, *CAR*, $\sigma(ROA)$) of Z-scores. Table 4 reports the results.

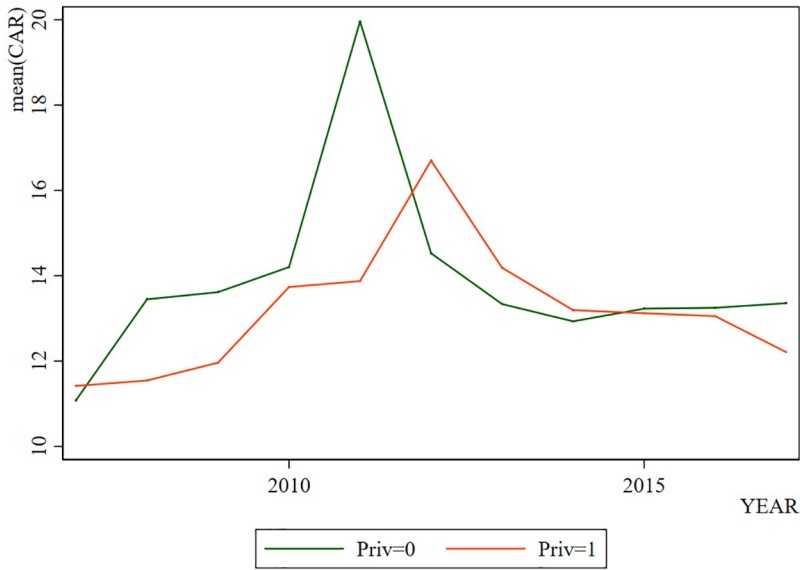

**Fig 3.** *CAR* **in different years.**

**Table 3. Stability test of variables.** This table reports the result of stability test according to the Fisher-ADF test.

| Variables | Statistics (Inverse Chi-squared P) | P-value |
|---|---|---|
| Ln_Z | 531.934 | 0.0000 |
| Priv | 276.642 | 0.0014 |
| Finawa | 1237.008 | 0.0000 |
| Loan | 430.687 | 0.0000 |
| NSCR | 456.228 | 0.0000 |
| NPL | 1000.101 | 0.0000 |
| CAR | 1007.273 | 0.0000 |
| GDP | 702.057 | 0.0000 |

In order to prove that the model setting and the system GMM estimation are reasonable, we first run the fixed-effect model regression, and the results are shown in column 1. It shows that the effect of *Priv* is not significant, indicating that the fixed effect model cannot effectively estimate the impact of private capital on the stability of banks. Column 2 shows that the impact of *Priv* is positive and significant. This indicates that private capital holding can significantly improve the stability of banks. Furthermore, the effects of *Priv* in Columns 3—5 are positive, negative, and negative, respectively, and significant. This shows that private capital holding can improve the stability of banks, mainly by improving the profitability of banks and reducing

**Table 4. Private capital holding, government's financial awareness and stability of CCBs.** This table reports the results of dynamic panel system GMM regressions analyzing the impact of private capital holding on stability of CCBs and the moderating effect of government's financial awareness (The column 1 is the estimated results of the fixed effect model). The dependent variables of Columns 1- 5 are $Ln\_Z$, $Ln\_Z$, $ROA$, $CAR$ and $\sigma(ROA)$ respectively. Variables are defined in Table 1. The sample period is from 2007 to 2017. In parentheses are t-statistics. ***, **, * indicate statistical significance at the 1%, 5%, and 10% level, respectively.

| | (1) Ln_Z | (2) Ln_Z | (3) ROA | (4) CAR | (5) Volatility of ROA |
|---|---|---|---|---|---|
| Priv | 0.792 | 1.552*** | 0.531*** | -0.178*** | -0.224*** |
| | (1.00) | (5.94) | (2.97) | (-7.45) | (-3.31) |
| Finawa | 1.052** | -0.089 | -0.141 | 0.003 | -0.155*** |
| | (2.21) | (-0.39) | (-1.26) | (0.50) | (-3.71) |
| Priv × Finawa | -1.247 | -1.700*** | -0.499** | 0.223*** | 0.184* |
| | (-1.08) | (-4.06) | (-2.23) | (7.10) | (1.87) |
| $Ln\_Z_{t-1}$ | | 0.802*** | | | |
| | | (20.47) | | | |
| $ROA_{t-1}$ | | | 0.422*** | | |
| | | | (18.34) | | |
| $CAR_{t-1}$ | | | | 0.167*** | |
| | | | | (20.80) | |
| $\sigma(ROA)_{t-1}$ | | | | | 0.400*** |
| | | | | | (14.28) |
| bank individual effect/year effect/X | Yes | Yes | Yes | Yes | Yes |
| N | 530 | 466 | 658 | 681 | 480 |
| Wald | | 2930.05 | 1998.45 | 715.49 | 1199.29 |
| | | 0.0000 | 0.0000 | 0.0000 | 0.0000 |
| Ar(1)-p | | 0.0000 | 0.0097 | 0.0518 | 0.0001 |
| Ar(2)-p | | 0.4549 | 0.3031 | 0.5728 | 0.3883 |
| Sargan-p | | 0.3839 | 0.2039 | 0.1907 | 0.3812 |
| F | 6.753 | | | | |

the volatility of profits. Here, research **Hypothesis 1** is confirmed. Private capital has higher operating efficiency and its own profit-seeking nature, which makes private shareholders pay more attention to the profitability of banks. This has led to better earnings performance of privately held banks, thus improving bank stability.

Importantly, We test the moderating effect of government financial awareness by interacting *Priv* with *Finawa*. The coefficient of this interaction term is our focus. Similarly, the interaction term are not significant in column 1, indicating that the fixed effect model still cannot effectively estimate the moderating effect. In column 2, the coefficient of interaction term is significantly negative. This implies that when the government's financial awareness is weak, private capital holding is not conducive to the stability of banks. The coefficients of interaction term in columns 3 to 5 are negative, positive and positive respectively, and significant. It shows that the negative moderating effect of financial awareness mainly reduces the profitability of banks and improves the volatility of profits. Here, research **Hypothesis 2** is confirmed.

We need to carry out rigorous tests, especially the autocorrelation test of the random disturbance term and the over-identification test, to identify that the dynamic panel system GMM regression is reasonable. In the Arellano-Bond test, the difference of random disturbance terms shows that all Ar(1)-p is less than 0.1, and all Ar(2)-p is greater than 0.1, that is, the difference of random disturbance terms has first-order autocorrelation and no second-order autocorrelation. The Sargan test of overidentifying restrictions shows that all Sargan-p are greater than 0.1, this means that the model setting passes the over-identification test. In addition, the coefficient of the lag term is significantly positive. Therefore, the dynamic panel system GMM regression was proved to be reasonable.

## Robustness checks

In this section, based on the shares of the top ten shareholders of the bank, we respectively build: (1) *spriv*, the sum of private ownership shares among the top ten shareholders; (2) *Privl*, a dummy variable, if the sum of private ownership shares among the top ten shareholders is greater than the sum of state-owned ownership shares, the value is 1; otherwise, it is 0. These two indicators can measure the controlling position of private capital in the ownership structure of banks to a certain extent.

### Use *spriv* as the independent variable

We first use *spriv* for robustness test. Table 5 reports the regression results of dynamic panel systems GMM. It shows that the effects of spriv and *spriv × Finawa* are basically consistent with Table 4. Our results are therefore more credible.

### Use *Privl* as the independent variable

Next, we use *Privl* to replace *Priv* for robustness testing. Table 6 reports the regression results of dynamic panel systems GMM. It shows that the effects of *Privl* and *Privl × Finawa* are still basically consistent with Table 4. This again provides reliable regression results.

## Endogeneity explanation

The dynamic panel model may still has the predetermined endogenous of private capital holding because the possible mutual causality relation between private capital holding and *ROA*, capital adequacy ratio and Z-score. Therefore, We further use the two-stage least square method of instrumental variables for robustness test.

**Table 5. Private ownership shares (*spriv*), government's financial awareness of and stability of CCBs.** This table reports the results of dynamic panel system GMM regressions analyzing the impact of private ownership shares (*spriv*) on stability of CCBs and the moderating effect of government's financial awareness. The dependent variables of Columns 1- 4 are *Ln_Z*, *ROA*, *CAR* and *σ(ROA)* respectively. Variables are defined in Table 1. The sample period is from 2007 to 2017. In parentheses are t-statistics. ***, **, * indicate statistical significance at the 1%, 5%, and 10% level, respectively.

| | (1) *Ln_Z* | (2) *ROA* | (3) *CAR* | (4) Volatility of *ROA* |
|---|---|---|---|---|
| *spriv* | 0.099*** | 0.036*** | -0.006*** | -0.028*** |
| | (3.02) | (3.02) | (-3.04) | (-3.37) |
| *Finawa* | 0.803*** | -0.392*** | 0.028*** | -0.145** |
| | (3.22) | (-4.86) | (4.65) | (-2.48) |
| *spriv × Finawa* | -0.165*** | -0.049** | 0.011*** | 0.055*** |
| | (-2.86) | (-2.36) | (3.94) | (3.49) |
| $Ln\_Z_{t-1}$ | 0.762*** | | | |
| | (29.25) | | | |
| $ROA_{t-1}$ | | 0.468*** | | |
| | | (27.88) | | |
| $CAR_{t-1}$ | | | 0.312*** | |
| | | | (29.61) | |
| $\sigma(ROA)_{t-1}$ | | | | 0.741*** |
| | | | | (18.05) |
| bank individual effect/year effect/*X* | Yes | Yes | Yes | Yes |
| N | 401 | 538 | 542 | 411 |
| Wald | 2644.48 | 2907.22 | 2030.6 | 956.38 |
| | 0.0000 | 0.0000 | 0.0000 | 0.0000 |
| Ar(1)-p | 0.0000 | 0.0024 | 0.0031 | 0.0016 |
| Ar(2)-p | 0.9910 | 0.1249 | 0.3229 | 0.1689 |
| Sargan-p | 0.8027 | 0.3153 | 0.2902 | 0.3341 |

The instrumental variable to be constructed is the weighted average value of private capital holding assets of CCBs in the same province. The corresponding private capital holding instrument variable of each CCB is the average value calculated by weighted total asset size according to the private capital holding situation of the remaining CCBs in the same province after excluding the private capital holding of this bank, so as to make the index more clean. There are two reasons why the weighted average value of private capital holding of CCBs in the same province (peer-average) can become an appropriate tool variable to measure private capital holding of each CCB in that year. First, the ownership structure of a CCB is often similar to the average situation of other CCBs in the same province. Thus, the instrumental variable has a strong correlation with the private capital holding situation of each CCB, so that overcome the problem of weak instrumental variable. Second, the average level of private capital holding of CCBs in the same province has a certain exogenous effect on the performance, risk and robust operation of individual CCB as the performance, risk and robustness of individual CCB are more determined by itself. This kind of method is also widely used to solve endogenous problems in similar literatures [5, 23, 24]. Finally, in order to solve the endogeneity more thoroughly, we use the lagged values of the peer-average measure in the regression.

We separately select and calculate whether the largest shareholder of CCBs is private capital to measure the private capital holding (*Priv*), the sum of the proportion of private equity in the top ten shareholders to measure private capital holding (*spriv*) and whether private equity in the top ten shareholders is greater than state-owned equity to measure private capital holding (*Privl*). The two-stage least square method is used to test the robustness.

**Table 6. Private capital relative holding (*Privl*), government's financial awareness of and stability of CCBs.** This table reports the results of dynamic panel system GMM regressions analyzing the impact of private capital relative holding (*Privl*) on stability of CCBs and the moderating effect of government's financial awareness. The dependent variables of Columns 1- 4 are *Ln_Z*, *ROA*, *CAR* and *σ(ROA)* respectively. Variables are defined in Table 1. The sample period is from 2007 to 2017. In parentheses are t-statistics. \*\*\*, \*\*, \* indicate statistical significance at the 1%, 5%, and 10% level, respectively.

| | (1) *Ln_Z* | (2) *ROA* | (3) *CAR* | (4) **Volatility of *ROA*** |
|---|---|---|---|---|
| *Privl* | 0.678** | 0.103*** | -0.013*** | -0.120** |
| | (2.48) | (2.74) | (-3.00) | (-2.35) |
| *Finawa* | 0.390 | -0.353*** | 0.031*** | -0.129** |
| | (1.46) | (-5.32) | (4.68) | (-2.19) |
| *Privl × Finawa* | -1.093** | -0.240*** | 0.024*** | 0.181** |
| | (-2.29) | (-3.40) | (2.95) | (2.13) |
| $Ln\_Z_{t-1}$ | 0.834*** | | | |
| | (25.63) | | | |
| $ROA_{t-1}$ | | 0.533*** | | |
| | | (26.18) | | |
| $CAR_{t-1}$ | | | 0.300*** | |
| | | | (27.06) | |
| $\sigma(ROA)_{t-1}$ | | | | 0.607*** |
| | | | | (23.30) |
| bank individual effect/year effect/*X* | Yes | Yes | Yes | Yes |
| N | 436 | 586 | 590 | 448 |
| Wald | 3595.37 | 2153.18 | 1729.6 | 1375.12 |
| | 0.0000 | 0.0000 | 0.0000 | 0.0000 |
| Ar(1)-p | 0.0000 | 0.0008 | 0.0009 | 0.0006 |
| Ar(2)-p | 0.7330 | 0.0813 | 0.6139 | 0.1257 |
| Sargan-p | 0.6700 | 0.2557 | 0.1594 | 0.4125 |

Tables 7–9 report the estimation results of two-stage least squares regression of the instrumental variable of *Priv*, *spriv* and *Privl* respectively. In these tables, Columns 1 and 2 are the results of the first stage, Columns 3 to 6 are the results of the second stage. These show that the effects of *Priv* and *Priv × Finawa* (*spriv* and *spriv × Finawa*, *Privl* and *Privl × Finawa*) are still basically consistent with Table 4. This again shows that the empirical results are robust and reliable.

## Conclusion and enlightenment

### Conclusion

Through rigorous empirical analysis, we draw the following conclusions: First, private capital holding is conducive to improving the operating performance of city commercial banks, reducing the volatility of their operating performance, which is conducive to the stability of city commercial banks. This is because private capital holding can ease the administrative intervention of local governments on city commercial banks' loans. This reduces the loans of city commercial banks to inefficient state-owned enterprises and makes it more convenient for city commercial banks to choose appropriate loan objects in a market-oriented way, so as to improve the efficiency of financial resource allocation. Second, the weaker the financial awareness of local governments, the worse the operating performance of city commercial banks held by private capital, which is ultimately not conducive to the stability of city commercial banks. The government gradually gives up its influence on local financial resources, which will also lead to the lack of financial supervision by local governments, which may eventually lead to the

**Table 7. *Priv*'s instrumental variable regression.** This table reports the results of 2SLS regressions analyzing the impact of private capital holding (*Priv*) on stability of CCBs and the moderating effect of government's financial awareness. The dependent variables of Columns 1- 6 are *Priv*, *Priv* × *Finawa*, *Ln_Z*, *ROA*, *CAR* and *σ(ROA)* respectively. Variables are defined in Table 1. The sample period is from 2007 to 2017. In parentheses are t-statistics. ***, **, * indicate statistical significance at the 1%, 5%, and 10% level, respectively.

| | First stage | | Second stage | | | |
|---|---|---|---|---|---|---|
| | **(1) *Priv*** | **(2) *Priv* × *Finawa*** | **(3) *Ln_Z*** | **(4) *ROA*** | **(5) *CAR*** | **(6) Volatility of *ROA*** |
| *Priv_iv* | 0.499* | 0.455*** | | | | |
| | (2.22) | (3.14) | | | | |
| *Priv_iv* × *Finawa* | | 1.082*** | | | | |
| | | (4.19) | | | | |
| *Priv* | | | 13.030** | 10.020** | -1.859* | -1.913** |
| | | | (2.12) | (2.10) | (-1.89) | (-2.05) |
| *Finawa* | 0.634*** | 0.631*** | -1.046** | -0.686 | -0.093 | -0.011 |
| | (5.87) | (9.07) | (-1.97) | (-1.35) | (-0.93) | (-0.06) |
| *Priv* × *Finawa* | | | -14.460* | -10.030* | 1.954* | 1.876* |
| | | | (-1.90) | (-1.67) | (1.75) | (1.72) |
| bank individual effect/year effect/*X* | Yes | Yes | Yes | Yes | Yes | Yes |
| N | 694 | 694 | 530 | 650 | 680 | 469 |

abuse of control power of private capital and damage the steady operation of city commercial banks.

We provide an in-depth understanding of the impact of private shareholders on the operation and management of banks. Further, we have enriched the theory of corporate governance of commercial banks. In addition, we have created unique insights into the role of government. Our research can provide the following enlightenment. First: It is of positive significance to introduce private capital to hold bank ownership, but in the process of introducing private capital, we should focus on whether private capital can become the controlling shareholder of city commercial banks. Because in the case of insufficient local government supervision, private capital has more power to transfer benefits to related enterprises, and the risk is ultimately

**Table 8. *spriv*'s instrumental variable regression.** This table reports the results of 2SLS regressions analyzing the impact of private capital holding (*spriv*) on stability of CCBs and the moderating effect of government's financial awareness. The dependent variables of Columns 1- 6 are *spriv*, *spriv* × *Finawa*, *Ln_Z*, *ROA*, *CAR* and *σ(ROA)* respectively. Variables are defined in Table 1. The sample period is from 2007 to 2017. In parentheses are t-statistics. ***, **, * indicate statistical significance at the 1%, 5%, and 10% level, respectively.

| | First stage | | Second stage | | | |
|---|---|---|---|---|---|---|
| | **(1) *spriv*** | **(2) *spriv* × *Finawa*** | **(3) *Ln_Z*** | **(4) *ROA*** | **(5) *CAR*** | **(6) Volatility of *ROA*** |
| *spriv _iv* | 0.919*** | 0.571*** | | | | |
| | (10.15) | (8.36) | | | | |
| *spriv_iv* × *Finawa* | | 0.172*** | | | | |
| | | (2.75) | | | | |
| *spriv* | | | 2.449** | 0.598*** | -0.017* | -1.356** |
| | | | (1.99) | (3.02) | (-1.89) | (-2.36) |
| *Finawa* | 0.484 | 1.126*** | 5.893** | 1.815*** | -0.016** | -2.352* |
| | (1.31) | (4.34) | (2.35) | (4.55) | (-2.11) | (-1.88) |
| *spriv* × *Finawa* | | | -3.824* | -0.936*** | 0.026** | 1.922* |
| | | | (-1.93) | (-2.94) | (2.03) | (1.95) |
| bank individual effect/year effect/*X* | Yes | Yes | Yes | Yes | Yes | Yes |
| N | 472 | 472 | 387 | 467 | 466 | 387 |

**Table 9. *Privl*'s instrumental variable regression.** This table reports the results of 2SLS regressions analyzing the impact of private capital holding (*Privl*) on stability of CCBs and the moderating effect of government's financial awareness. The dependent variables of Columns 1- 6 are *Privl*, *Privl × Finawa*, *Ln_Z*, *ROA*, *CAR* and *σ(ROA)* respectively. Variables are defined in Table 1. The sample period is from 2007 to 2017. In parentheses are t-statistics. ***, **, * indicate statistical significance at the 1%, 5%, and 10% level, respectively.

| | First stage | | Second stage | | | |
|---|---|---|---|---|---|---|
| | **(1) Privl** | **(2) Privl × Finawa** | **(3) Ln_Z** | **(4) ROA** | **(5) CAR** | **(6) Volatility of ROA** |
| *Privl_iv* | 0.478** | 0.0857 | | | | |
| | (2.08) | (0.61) | | | | |
| *Privl_iv × Finawa* | | 0.308** | | | | |
| | | (2.34) | | | | |
| *Privl* | | | 2.046** | 0.570* | -0.158* | -0.968*** |
| | | | (2.20) | (1.65) | (-1.89) | (-4.38) |
| *Finawa* | 1.374*** | 1.052*** | 0.368 | 0.776* | -0.285*** | -0.188 |
| | (9.64) | (11.98) | (0.47) | (1.78) | (-3.80) | (-0.89) |
| *Privl × Finawa* | | | -3.323** | -1.389** | 0.474*** | 1.451*** |
| | | | (-2.13) | (-2.20) | (3.15) | (3.59) |
| bank individual effect/year effect/X | Yes | Yes | Yes | Yes | Yes | Yes |
| N | 595 | 595 | 477 | 589 | 581 | 435 |

borne by banks. Therefore, private capital should not be the actual controller of city commercial banks when they have not formed good external governance. Second: As an important relationship person of city commercial banks, local governments should not completely withdraw. Local governments should actively play a supervisory role (shareholder supervision and administrative supervision), prevent potential benefit transfer, promote the steady operation of city commercial banks, and maintain local financial stability to avoid systemic financial risks.

## Future prospects

Our research still has certain limitations. We have not analyzed the impact of private capital holdings on bank stability from the perspective of corporate governance mechanism. Future research can deeply study the topic of this paper from the perspective of corporate governance (including equity checks and balances, principal-agent, executive incentives, etc.). Of course, the impact of private capital holding banks is more extensive, such as bank loans, risk taking and social responsibility. We can also study the moderating effect of macro policies (monetary policy, fiscal policy, industrial policy, etc.) on the role of private capital.

## Supporting information

**S1 Data.**
(XLSX)

## Author Contributions

**Conceptualization:** Jie Liu, Jiarong Li.

**Data curation:** Jie Liu.

**Formal analysis:** Kun Xu, Jie Liu.

**Investigation:** Kun Xu, Jie Liu.

**Methodology:** Kun Xu, Jie Liu.

**Software:** Jie Liu.

**Validation:** Jie Liu.

**Visualization:** Jie Liu.

**Writing – original draft:** Jie Liu.

**Writing – review & editing:** Kun Xu, Jie Liu, Jiarong Li.

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
