## [Decision Letter · Decision Letter 0]

8 Jan 2023

PONE-D-22-34925Private capital holding, financial awareness of government and steadying operation of banks: Evidence from ChinaPLOS ONE

Dear Dr. Jie Liu,

Thank you for submitting your manuscript to PLOS ONE. After careful consideration, we feel that it has merit but does not fully meet PLOS ONE’s publication criteria as it currently stands. Therefore, we invite you to submit a revised version of the manuscript that addresses the points raised during the review process.

We look forward to receiving your revised manuscript.

Kind regards,

María del Carmen Valls Martínez, Ph.D.

Academic Editor

PLOS ONE

Journal Requirements:

"Kun Xu’s research was funded by Humanities and Social Sciences Foundation of the Ministry of Education of China (No. 19YJC790162)."

Reviewers' comments:

Reviewer's Responses to Questions

**Comments to the Author**

1. Is the manuscript technically sound, and do the data support the conclusions?

Reviewer #1: Yes

Reviewer #2: Partly

Reviewer #3: Partly

Reviewer #4: Yes

2. Has the statistical analysis been performed appropriately and rigorously? 

Reviewer #1: Yes

Reviewer #2: Yes

Reviewer #3: Yes

Reviewer #4: Yes

3. Have the authors made all data underlying the findings in their manuscript fully available?

Reviewer #1: Yes

Reviewer #2: No

Reviewer #3: Yes

Reviewer #4: Yes

4. Is the manuscript presented in an intelligible fashion and written in standard English?

Reviewer #1: Yes

Reviewer #2: Yes

Reviewer #3: No

Reviewer #4: Yes

5. Review Comments to the Author

Reviewer #1: This is a great work. The paper is clear and well-written on the topic of bank behaviour in China. The approach taken is appropriate to answering the question posed by the authors. I think it is ready for publication.

Reviewer #2: Review comments: major revision

Question 1: 123 city construction banks are used as individuals in the panel data. Then how is the data of “Finawa” (the government's financial awareness) collected and obtained? The author said it was obtained from the annual reports of 5 financial institutions: banks, insurances, securities, funds and trusts. Are these 5 financial institutions related to 123 city China Construction banks? Please explain. Similarly, how does the author relate GDP to individual city construction banks? Please answer. Does the reviewer mean that this paper uses bank-city-level panel data or bank-level panel data?

Question 2: If it is bank-city level panel data, the data of GDP should be specific to city level instead of provincial level panel data obtained by National Bureau of Statistics. Please kindly find data from “China City Statistical Yearbook” for matching (according to the reviewer, this part of data should be found).

Suggestion 1: Since “Financial Awareness of Government” (Finawa) works on “the steady operation of banks” (ln-Z) through “Private capital holding” (Priv), That is, it (Finawa) plays an intermediary role, so in subsection 2 of "2Literature review and research hypothesis", It should not be "Financial awareness of government and steady operation of banks", please correct the title of subsubsection 2., modify Hypothesis 2 and modify the theoretical analysis.

Suggestion 2: SYS-GMM method was adopted in this paper to mainly solve the endogeneity of explained variables and lagged explained variables, which could not solve the problem of omission of important variables. Adding control variables could solve the endogeneity problem caused by missing variables. Therefore, please correct this statement. In addition, control variables are also included in this article. Please explain there that model endogeneity problems caused by missing variables can be solved.

Suggestion 3: In "measures", the author says that the time trend item YEAR is added to control the time effect, but it is not included in the "model expression" (YEAR is not included in the control variable). Please add it in order to make the model expression more complete.

Suggestion 4: SYS-GMM is used in this paper, so whether the applicable conditions of SYS-GMM are applicable should be explained. Please kindly give a supplementary explanation (such as: automatic correlation test of random disturbance term, over-recognition test).

Suggestion 5: Table A1.Variable definitions should appear in the text, not in the appendix. Only in this way can readers understand. It is suggested to add "3 Data and methods", and add a subsection of "Variable Description", in which details are explained: 1) explained variables;2) Core explanatory variable;3) Control variables, and attach “Table A1.Variable definitions”, so that readers have a detailed understanding of variable selection.(For example, how the explained variable ln_Z was obtained, although the author explains a little in the "results" section, please explain in the "variable description" section.)

Suggestion 6: Please add a subsection of "Stability test of variable data" to "4Results and discussion" to make the analysis of empirical results more complete (although the short panel data is basically stable).

Suggestion 7: In general, the endogeneity problem is basically solved using the SYS-GMM method. However, the author also mentioned "the endogeneity problem that the core explanatory variable and the explained variable may be causal to each other". Since this is taken into consideration, please kindly add a part of "6 Endogeneity explanation", and please elaborate in this part.

Suggestion 8: In "5 Robustness checks", the method of replacing core explanatory variables (two replacement variables are found) is used to test robustness. Please kindly ask the author to explain it in two subsections, so that readers can be clearer.

Suggestion 9: "Note" in Table2-Table5 is usually placed under the table, please correct them.

Suggestion 10: A subsection "Future prospects" should be added to "Conclusions and Enlightenments", that is, "Future prospects" should be included.

Suggestion 11: Please be careful to correct translation errors (for example, 1)it is better to use the present tense in the "abstract" rather than the past tense; 2) Line 28: We attempts to illustrate that the introduction of private capital is conducive to the release of the potential efficiency of the CCBs, but the result is closely linked with the financial awareness of the government.; 3) Line 34: we sets up the index of government financial awareness based on the data of the top ten shareholders of 123 CCBs from 2007 to 2017 in China. etc.) and standardize the reference format of the article.

Reviewer #3: Reviewer Report on “Private capital holding, financial awareness of government and steadying operation of banks: Evidence from China.”

The paper under review examines two related questions: (1) whether (and how) private capital holding impacts the stability of bank operations in China. And (2) if so, does government awareness play a moderating role? A panel dataset on 123 city commercial banks (CCBs) was analyzed by a dynamic system GMM model. While I think the chosen topic is of both academic interest and policy relevance, the data and the modeling device are both suitable to answer the research questions, I also think there is much room for improvement in the paper. More specific comments are outlined below, largely by “section.”

More information should be included in the “Data” section.

1. How were the 123 CCBs chosen? Are they representative of the underly population of CCBs in China?

2. Why chose 2007-2017 as the study period? My concern is that this time period involves a time period for the global crisis. I am not sure how that event affects your key findings (--In particular, since the dynamic GMM model uses lagged variables and their differences as instruments for the endogenous variable, some of these lagged variables might have been contaminated with the global crisis). In fact, this issue should be checked in the “robustness checks” section.

3. There should be some paragraphs in the “Data” section to talk about your variables. For example, how are “stability” and “government awareness” measured? For example, how may a read understand what 1.944, the mean of “Ln_Z,” means? And why should it be taken logarithm? Yes, there are some explanations provided in Table A1, but I think variable definition and construction deserve a more formal place in the text to better inform the reader.

4. Table A1 can be incorporated into Table 1.

5. The first paragraph on page 6 (on what is included in the Z-scores) should be put in the Data or Model section.

6. Since you have data on the time dimension, I think a graph depicting the time trends of the key variables would be very helpful for understanding your story.

Modeling/results:

1. The key explanatory variable, Priv_it is treated as a dummy variable. I wonder if it can be constructed as a continuous variable, i.e., the “share” of private capital. That way, more variation can be exploited in the regression.

2. Some of the explanatory variables are measured by their current values; some others are measured by their lagged values. More explanations should be provided to clear out the confusion. In fact, more discussions on why you selected your covariates (e.g., from theory or from the literature) may be added to the “Model” section.

3. Other than simply saying that “a panel system GMM” model is estimated, it would be nice to provide some discussion on how this model achieves identification (—and why it is chosen over conventional panel data models?).

4. In fact, it would also be informative to compare results based on conventional panel-data models, such as fixed effects and random effects models, with those of the dynamic GMM models to see if the dynamic GMM indeed has a significant advantage over the conventional models.

5. The panel seems to be missing a lot of observations for some variables, as shown in Table 1. This might introduce a sample selection issue to the model—for example, note that column 1 of Table 2 has only 466 obs which is only 2/3 of the sample size in column 2. I wonder how this issue was addressed.

6. In robustness checks, the authors use the “regional average” of the endogenous variable as an instrumental variable for it. This might be inappropriate. This strategy exploits “peer effects” for identification, but note that any bank is also a peer for other banks – so the influence goes both way, which creates a reverse causality issue in the first stage regression. A better strategy would be to use lagged values of the peer-average measure to create instrument variables.

The writing needs careful revisions. There are many typos, grammatical errors, and style issues in the text. For example,

(1) Abstract: what is ROA? As a general rule, acronyms should be spelled out at their first appearance.

(2) The directions of many of the quotation marks are wrong.

(3) Lines 105-109: Please check the sentence. It reads very confusingly.

(4) Line 160 (page 5/21): “…a dummy variables, …” should be “… a dummy variable, …”

(5) Line 181: “onstability” should be “on stability.”

Reviewer #4: The paper deals with the relationship between private capital holding and steadying operation of banks, which can be very essential for the regulatory discussions and financial stability -in general- especially in a developing country setting like China, and thus is highly relevant. In addition, the authors examined the impact of government financial awareness on the impact of private capital holdings on bank stability. The structure of the manuscript is clear. The empirical evidence is rigorous, the method is properly used, and a series of robustness tests are carried out to ensure that the results are reliable. The paper makes an important contribution to the body of knowledge. This is one of the few literatures that systematically study the influence of private capital holding on bank stability. Overall, I suggest publishing it after minor revision. Specific comments are as follows:

(1) The paper has made great contributions to the body of knowledge, but in the last part of the introduction, the author's summary of contributions is too brief and unclear. I suggest reorganizing this part to highlight the significance.

(2) In the results section, although the regression results confirm research hypothesis 1 and research hypothesis 2, I still recommend briefly summarizing research hypothesis 1 and research hypothesis 2 in this section. In addition, the authors also need to add appropriate economic meanings to explain what causes these results. This will make it easier for readers to understand.

(3) The table of regression results is too large. I suggest leaving out unimportant control variables and constant items and replacing them with YES or NO.

(4) Add contribution for body of knowledge in Conclusion section.

(5) Add limitation and suggestion for future research in Conclusion section.

6. PLOS authors have the option to publish the peer review history of their article (what does this mean?). If published, this will include your full peer review and any attached files.

Reviewer #1: No

Reviewer #2: **Yes: **Haisong Dong

Reviewer #3: No

Reviewer #4: No

---

## [Author Response · Author response to Decision Letter 0]

21 Jan 2023

Response to reviewer #1 comments

Dear Reviewer #1,

We are very grateful for your appreciation of our paper, which makes us very encouraged.

Response to reviewer #2 comments

Dear Reviewer Haisong Dong,

We really appreciate you for your carefulness and conscientiousness. Your suggestions are really valuable and helpful for revising and improving our paper. According to your suggestions, we have made the following revisions on this manuscript: 

Question 1: 123 city construction banks are used as individuals in the panel data. Then how is the data of “Finawa” (the government's financial awareness) collected and obtained? The author said it was obtained from the annual reports of 5 financial institutions: banks, insurances, securities, funds and trusts. Are these 5 financial institutions related to 123 city China Construction banks? Please explain. Similarly, how does the author relate GDP to individual city construction banks? Please answer. Does the reviewer mean that this paper uses bank-city-level panel data or bank-level panel data?

Response 1：

First, the financial awareness of local governments (Finawa) is a provincial indicator.

In China, major financial institutions include banks, insurance institutions, securities, funds and trusts. According to the annual reports of various financial institutions in various provinces, we collect the information of ownership structure of each institution to obtain the amount of shares held by the local government, and calculate the assets held by the local government. Of course, we also need to add up the total assets of all financial institutions in the province to calculate the average share of financial assets held by local governments, namely, Finawa. The larger the share, the stronger the financial awareness of local governments and the more financial assets they hold.

Of course, these five types of financial institutions include local city commercial banks, but the difference is that Finawa is a provincial indicator, while the financial data of city commercial banks are bank-level indicators.

Finally, we should note that our data structure is the panel data at the bank level. Although the Finawa is a provincial indicator, it eventually matches each bank.

Question 2: If it is bank-city level panel data, the data of GDP should be specific to city level instead of provincial level panel data obtained by National Bureau of Statistics. Please kindly find data from “China City Statistical Yearbook” for matching (according to the reviewer, this part of data should be found).

Response 2：

This is our negligence. In fact, in China, the operation of city commercial banks is not limited to local cities, but all important cities in the province where they are located. This is also the reason why we choose the GDP of the province where the city commercial banks are located instead of the city GDP. We have added an explanation in the Measures section.

Suggestion 1: Since “Financial Awareness of Government” (Finawa) works on “the steady operation of banks” (ln-Z) through “Private capital holding” (Priv), That is, it (Finawa) plays an intermediary role, so in subsection 2 of "2Literature review and research hypothesis", It should not be "Financial awareness of government and steady operation of banks", please correct the title of subsubsection 2., modify Hypothesis 2 and modify the theoretical analysis.

Response 1：

We need to note that the government's financial awareness (Finawa) does not affect the steady operation of banks (Ln_Z) through private capital holding (Priv), that is, it is not an intermediary role. What we are talking about is the moderating effect. Based on the analysis of the impact of Priv on Ln_Z, we further studied whether Finawa changed the impact of Priv.

Suggestion 2: SYS-GMM method was adopted in this paper to mainly solve the endogeneity of explained variables and lagged explained variables, which could not solve the problem of omission of important variables. Adding control variables could solve the endogeneity problem caused by missing variables. Therefore, please correct this statement. In addition, control variables are also included in this article. Please explain there that model endogeneity problems caused by missing variables can be solved.

Response 2：

We have corrected the relevant statement and explained the role of adding control variables according to your suggestion.

Suggestion 3: In "measures", the author says that the time trend item YEAR is added to control the time effect, but it is not included in the "model expression" (YEAR is not included in the control variable). Please add it in order to make the model expression more complete.

Response 3：

We have added the variable YEAR to the model.

Suggestion 4: SYS-GMM is used in this paper, so whether the applicable conditions of SYS-GMM are applicable should be explained. Please kindly give a supplementary explanation (such as: automatic correlation test of random disturbance term, over-recognition test).

Response 4：

In the original manuscript, we simply explained that the model was proved to be reasonable through relevant tests. However, we have not specified which tests have been passed. We have revised them according to your suggestion.

Suggestion 5: Table A1.Variable definitions should appear in the text, not in the appendix. Only in this way can readers understand. It is suggested to add "3 Data and methods", and add a subsection of "Variable Description", in which details are explained: 1) explained variables;2) Core explanatory variable;3) Control variables, and attach “Table A1.Variable definitions”, so that readers have a detailed understanding of variable selection.(For example, how the explained variable ln_Z was obtained, although the author explains a little in the "results" section, please explain in the "variable description" section.)

Response 5：

We have added the subsection "Variable description" in the "Data and methods". Of course, we also attached a table of variable definitions in this subsection.

Suggestion 6: Please add a subsection of "Stability test of variable data" to "4Results and discussion" to make the analysis of empirical results more complete (although the short panel data is basically stable).

Response 6：

We added the stationarity test of variable data.

Suggestion 7: In general, the endogeneity problem is basically solved using the SYS-GMM method. However, the author also mentioned "the endogeneity problem that the core explanatory variable and the explained variable may be causal to each other". Since this is taken into consideration, please kindly add a part of "6 Endogeneity explanation", and please elaborate in this part.

Response 7：

We arranged the subsection of "Instrumental variable" in the original manuscript into a new section "Endogeneity explanation".

Suggestion 8: In "5 Robustness checks", the method of replacing core explanatory variables (two replacement variables are found) is used to test robustness. Please kindly ask the author to explain it in two subsections, so that readers can be clearer.

Response 8：

According to two different variables, we split the subsection of "Replace independent variable" into two subsections.

Suggestion 9: "Note" in Table2-Table5 is usually placed under the table, please correct them.

Response 9：

Indeed, the settings of "Note" for tables in different publications are different. We refer to similar articles published in PLOS ONE again (Liu et al., 2022a;2022b), and confirm the correct location of "Note".

Liu J, Xu L, Zhang Q. The influence of the largest private shareholder on bank loans: Evidence from China. PLoS ONE. 2022; 17(10): e0276877. https://doi.org/10.1371/journal.pone.0276877

Liu J, Zhang Q, Xiao C. Large private shareholders, industrial policies and industrial loans of city commercial banks: Evidence from China. PLoS ONE. 2022; 17(12): e0278654. https://doi.org/10.1371/journal.pone.0278654

Suggestion 10: A subsection "Future prospects" should be added to "Conclusions and Enlightenments", that is, "Future prospects" should be included.

Response 10：

We added a section "Future prospects" in "Conclusions and Enlightenments".

Suggestion 11: Please be careful to correct translation errors (for example, 1)it is better to use the present tense in the "abstract" rather than the past tense; 2) Line 28: We attempts to illustrate that the introduction of private capital is conducive to the release of the potential efficiency of the CCBs, but the result is closely linked with the financial awareness of the government.; 3) Line 34: we sets up the index of government financial awareness based on the data of the top ten shareholders of 123 CCBs from 2007 to 2017 in China. etc.) and standardize the reference format of the article.

Response 11：

We reviewed the grammar standards and related formats.

Response to reviewer #3 comments

Dear Reviewer #3,

We appreciate your recognition of our article. Your suggestions are very helpful to improve the quality of our paper. According to your suggestions, we have made the following revisions on this manuscript: 

More information should be included in the “Data” section.

1. How were the 123 CCBs chosen? Are they representative of the underly population of CCBs in China?

Response 1：

As of January 2023, there are a total of 125 city commercial banks in China (very few of them have come from the restructuring of our sample banks). The information disclosure of some banks is not perfect (the scale is also small), so we can't get their relevant information, and finally we got 123 banks. Therefore, our sample represents almost all city commercial banks in China.

2. Why chose 2007-2017 as the study period? My concern is that this time period involves a time period for the global crisis. I am not sure how that event affects your key findings (--In particular, since the dynamic GMM model uses lagged variables and their differences as instruments for the endogenous variable, some of these lagged variables might have been contaminated with the global crisis). In fact, this issue should be checked in the “robustness checks” section.

Response 2：

On the one hand, city commercial banks began to disclose annual reports generally and in a standard manner in 2007. On the other hand, since 2018, some new city commercial banks have been established by restructuring several city commercial banks. These lead to our sample period of 2007-2017. Of course, this period includes the global crisis. However, in the process of indicator construction (the moving average of the three periods), most of the samples during the crisis were eliminated. By selecting the sample after 2008, the regression results are shown in the following table. It can be found that the column 1 and column 4 are the same as the regression results in the manuscript, while column 2 and column 3 are almost the same.

 (1) (2) (3) (4)

 Ln_Z ROA CAR Volatility of ROA

Priv 1.5521*** 0.5642*** -0.1783*** -0.2240***

 (5.94) (3.04) (-7.48) (-3.31)

Finawa -0.0886 -0.1441 0.0024 -0.1548***

 (-0.39) (-1.30) (0.46) (-3.71)

Priv×Finawa -1.6996*** -0.5536** 0.2228*** 0.1844*

 (-4.06) (-2.36) (7.13) (1.87)

Ln_Zt-1 0.8021*** 

 (20.47) 

ROAt-1 0.4248*** 

 (18.17) 

CARt-1 0.1677*** 

 (20.88) 

σ(roa)t-1 0.3996***

 (14.28)

control variables/year Yes Yes Yes Yes

N 466 638 660 413

3. There should be some paragraphs in the “Data” section to talk about your variables. For example, how are “stability” and “government awareness” measured? For example, how may a read understand what 1.944, the mean of “Ln_Z,” means? And why should it be taken logarithm? Yes, there are some explanations provided in Table A1, but I think variable definition and construction deserve a more formal place in the text to better inform the reader.

4. Table A1 can be incorporated into Table 1.

5. The first paragraph on page 6 (on what is included in the Z-scores) should be put in the Data or Model section.

Response 3, 4 and 5：

We have added the subsection "Variable description" in the "Data and methods" to talk about these variables. Of course, we also attached a table of variable definitions (Table A1) in this subsection. In addition, we have moved the description of Z-scores (the first paragraph on page 6 of the original manuscript) to the "Variable description" subsection.

6. Since you have data on the time dimension, I think a graph depicting the time trends of the key variables would be very helpful for understanding your story.

Response 6：

We have added some graphs in the "Descriptive Statistics" subsection to show the changes of some key variables in the time dimension.

Modeling/results:

1. The key explanatory variable, Priv_it is treated as a dummy variable. I wonder if it can be constructed as a continuous variable, i.e., the “share” of private capital. That way, more variation can be exploited in the regression.

Response 1：

We did not choose the “share” because different banks have different degrees of equity dispersion. In extreme cases, the shareholding ratio of the largest shareholder may be close to 50%, but it may only be about 5%. Therefore, we pay more attention to the actual holding capacity of private shareholders, that is, whether they are the largest shareholder. Of course, in the "Robustness checks" section, we constructed the proportion of private shareholders to conduct a robustness test, so that more changes are used in the regression.

2. Some of the explanatory variables are measured by their current values; some others are measured by their lagged values. More explanations should be provided to clear out the confusion. In fact, more discussions on why you selected your covariates (e.g., from theory or from the literature) may be added to the “Model” section.

Response 2：

The addition of lag term is the original setting of system GMM estimation (Shen and Yao,2008; Che et al.,2013). This is to solve the inertia of the dependent variable, that is, the current operating conditions may depend on the past operating conditions. We have explained in more detail in the “Model specification” subsection.

Shen Y, Yao Y. Does grassroots democracy reduce income inequality in China? Journal of Public Economics. 2008;92(10-11): 2182-2198. DOI: 10.1016/j.jpubeco.2008.04.002.

Che Y, Lu Y, Tao Z, Wang P. The impact of income on democracy revisited. Journal of Comparative Economics. 2013; 41(1): 159-169. https://doi.org/10.1016/j.jce.2012.05.006.

3. Other than simply saying that “a panel system GMM” model is estimated, it would be nice to provide some discussion on how this model achieves identification (—and why it is chosen over conventional panel data models?).

Response 3：

In the “Model specification” subsection, we added an explanation of why the system GMM model is selected. In the “Results” section, we add the description of relevant identification tests.

4. In fact, it would also be informative to compare results based on conventional panel-data models, such as fixed effects and random effects models, with those of the dynamic GMM models to see if the dynamic GMM indeed has a significant advantage over the conventional models.

Response 4：

We added fixed effects regression in the “Results” section to compare with GMM estimation of dynamic panel system.

5. The panel seems to be missing a lot of observations for some variables, as shown in Table 1. This might introduce a sample selection issue to the model—for example, note that column 1 of Table 2 has only 466 obs which is only 2/3 of the sample size in column 2. I wonder how this issue was addressed.

Response 5：

This is caused by three reasons. First, our panel data is unbalanced. Some variables have more observations, while others have less. The second is that GMM estimation of dynamic panel system involves lag term, which will also cause the loss of some observations. The third is the construction of Ln_Z, which is calculated by the moving average of the three periods before and after ROA and CAR and the volatility of ROA, which also leads to the observations of Ln_Z being less than others.

6. In robustness checks, the authors use the “regional average” of the endogenous variable as an instrumental variable for it. This might be inappropriate. This strategy exploits “peer effects” for identification, but note that any bank is also a peer for other banks – so the influence goes both way, which creates a reverse causality issue in the first stage regression. A better strategy would be to use lagged values of the peer-average measure to create instrument variables.

Response 6：

We used the lag term of the peer-average to create the instrument variable for robustness checks.

The writing needs careful revisions. There are many typos, grammatical errors, and style issues in the text. For example,

(1) Abstract: what is ROA? As a general rule, acronyms should be spelled out at their first appearance.

(2) The directions of many of the quotation marks are wrong.

(3) Lines 105-109: Please check the sentence. It reads very confusingly.

(4) Line 160 (page 5/21): “…a dummy variables, …” should be “… a dummy variable, …”

(5) Line 181: “onstability” should be “on stability.”

Response：

We reviewed the spelling, grammar and style of the manuscript again and corrected the corresponding errors and deficiencies.

Response to reviewer #4 comments

Dear Reviewer #4,

We are very grateful for your favorable view on our manuscript. Your comments are very important to the acceptability of our paper. According to your suggestions, we have made the following revisions on this manuscript: 

(1) The paper has made great contributions to the body of knowledge, but in the last part of the introduction, the author's summary of contributions is too brief and unclear. I suggest reorganizing this part to highlight the significance.

Response 1：

In the last part of the “Introduction”, we rewrote our contributions and highlighted the importance of the paper.

(2) In the results section, although the regression results confirm research hypothesis 1 and research hypothesis 2, I still recommend briefly summarizing research hypothesis 1 and research hypothesis 2 in this section. In addition, the authors also need to add appropriate economic meanings to explain what causes these results. This will make it easier for readers to understand.

Response 2：

In the “Results” section, we further specify the research hypothesis 1 and hypothesis 2 in combination with the regression results. We have also made corresponding economic explanations for the results so that the author can better understand them.

(3) The table of regression results is too large. I suggest leaving out unimportant control variables and constant items and replacing them with YES or NO.

Response 3：

In the regression tables, we remove the unimportant control variables and constant terms and replace them with YES or NO.

(4) Add contribution for body of knowledge in Conclusion section.

Response 4：

We added our contribution to the body of knowledge in the “Conclusion”.

(5) Add limitation and suggestion for future research in Conclusion section.

Response 5：

We added limitation and suggestion for future research in the “Conclusion”.

---

## [Decision Letter · Decision Letter 1]

3 Feb 2023

PONE-D-22-34925R1Private capital holding, financial awareness of government and steadying operation of banks: Evidence from ChinaPLOS ONE

Dear Dr. Liu,

Thank you for submitting your manuscript to PLOS ONE. After careful consideration, we feel that it has merit but does not fully meet PLOS ONE’s publication criteria as it currently stands. Therefore, we invite you to submit a revised version of the manuscript that addresses the points raised during the review process.

We look forward to receiving your revised manuscript.

Kind regards,

María del Carmen Valls Martínez, Ph.D.

Academic Editor

PLOS ONE

Journal Requirements:

Reviewers' comments:

Reviewer's Responses to Questions

**Comments to the Author**

1. If the authors have adequately addressed your comments raised in a previous round of review and you feel that this manuscript is now acceptable for publication, you may indicate that here to bypass the “Comments to the Author” section, enter your conflict of interest statement in the “Confidential to Editor” section, and submit your "Accept" recommendation.

Reviewer #2: (No Response)

Reviewer #3: (No Response)

Reviewer #4: All comments have been addressed

2. Is the manuscript technically sound, and do the data support the conclusions?

Reviewer #2: Partly

Reviewer #3: Partly

Reviewer #4: Yes

3. Has the statistical analysis been performed appropriately and rigorously? 

Reviewer #2: No

Reviewer #3: Yes

Reviewer #4: Yes

4. Have the authors made all data underlying the findings in their manuscript fully available?

Reviewer #2: Yes

Reviewer #3: Yes

Reviewer #4: Yes

5. Is the manuscript presented in an intelligible fashion and written in standard English?

Reviewer #2: Yes

Reviewer #3: No

Reviewer #4: Yes

6. Review Comments to the Author

Reviewer #2: Review comments: major revision

Question 1: Among the 125 city commercial banks in China, the Bank of Guangzhou, the Bank of Tianjin, the Bank of Beijing, the Bank of Shanghai, the Bank of Qilu, the Bank of Nanjing, the Bank of Hangzhou, the Bank of Dalian, the Bank of Shengjing, the Bank of Changsha, the Bank of Chongqing, the Bank of Chengdu and the Bank of Xiamen, etc are all city commercial banks with branches outside the province. (For example, as of September 2011, Bank of Beijing has 9 branches in Shanghai, Tianjin, Hangzhou, Shenzhen, Xi 'an, Changsha, Nanjing, Jinan and Nanchang, covering 9 provinces in China.).Taking “GDP” as an example, according to the author's reply (since urban commercial banks have multiple branches in the province, provincial GDP is used instead of city-level GDP), the above banks must use cross-provincial GDP, even China's GDP. Similarly, “Finawa” (financial awareness of the government) cannot be a provincial index.

If the author must add macroeconomic variables, the reviewer provides several solutions. Method 1 is that the author takes the city where the headquarters of the city commercial bank is located as the reference and adopts city-level indicators. In this way, data must be found again from relevant databases such as China City Yearbook to recalculate "GDP" and "Finawa" (financial awareness of the government). And the text should have the corresponding explanation. Method 2 is to delete these inter-provincial city commercial banks and adopt provincial indicators according to the author's idea, but the sample size may be reduced or even insufficient. Please use your own discretion.

Of course, if the author has a more reasonable reason that the data need not be recollected or changed, the reviewer thinks that "minor revision" is OK, otherwise, it must be "major revision".

Suggestion 1: According to the author's reply to Suggestion 1 of Reviewer 2, the government's financial consciousness (Finawa) does not affect the stable operation of banks (Ln_Z) through private capital holding (Priv), that is, it is not an intermediary effect, but a moderating effect. On the basis of analyzing the influence of "Priv" on "Ln_Z", the reviewer further studies whether "Finawa" changes the influence of "Priv", which is agreed by the reviewer.

However, the title "Financial awareness of government and steady operation of banks" of the author's research hypothesis 2 and “research hypothesis 2” do not seem to reflect the moderating effect of "Finawa". Instead, introduce the direct effect of "Finawa" on the steady operation of banks (ln-Z). Therefore, the reviewer thinks it is necessary to revise the title of research hypothesis 2, research Hypothesis 2, and even the content analysis. Please modify it.

Suggestion 2: According to Suggestion 3 of Reviewer 2, the author added "YEAR" to “the model expression” to represent the year effect.

But please revise the wording. Line 184: "Of course, we also control the trend effect of year, YEAR." should be changed to "YEARt is year effect"; Line 178: "Wi is the individual fixed effect of the banks" is changed to "Wi is banks individual effect" and delete the word "fixed". Please modify it.

Suggestion 3: According to Suggestion 6 of Reviewer 2, the author adds a subsection "Stability test of Variable Data". But according to the author's modification, does “Fisher test” refer to “Fisher-ADF test” or “Fisher-PP test”? Please specify.

In Table 3, Column 1 should list the names of all variables, and the abbreviations of the variables before and after should be the same. For example, "P" should be changed to "Priv", "Z" should be changed to "Ln-Z", that is, to add logarithms, and so on. The reviewer even has the feeling that the author did not really test the Stability of the variables, which led to these errors. Please modify it.

Suggestion 4: According to Suggestion 4 of Reviewer 2, it is necessary to explain the application conditions of SYS-GMM method (such as autocorrelation test of random disturbance items and overrecognition test), but the reviewer found that the author did not explain in the paper (although there are relevant results in the table), please make a supplementary explanation.

The reviewer thinks this part is necessary for the following reasons: the empirical results of fixed effects of this paper show that the direct effect of "Priv" is not significant, and the moderating effect of "Finawa" is not significant, and the author wants to reach the research conclusion of this paper, so the SYS-GMM method is adopted. It must be explained here whether the application conditions of SYS-GMM method are met. In this way, the empirical results of SYS-GMM studied in this paper, “"Priv" has a direct effect and "Finawa" has a moderating effect”, are reasonable. Please modify it.

Suggestion 5: In "Table 4～Table 10", it is necessary to increase the "bank individual effect"; Time effect, should be "year effect", not "year"; The "control variables" should be changed to "X" (because "X" represents all control variables), and it is suggested that "X", "year effect" and "bank individual effect" should be displayed in three lines in "Table 4～Table 10", and the empirical results should be represented by "Yes". Please modify it.

Suggestion 6: In the "Future prospects" of this paper, please point out the limitations of this research, so as to lead to the future research direction of this topic. Please add clarification.

Reviewer #3: The revised draft has addressed most of my previous concerns. The one issue that remains is that of missing observations. While the authors explained why there were a large number of observations missing when they tried to perform robustness checks (using different models), they DID NOT ADDRESS the missing data issue--I am worried that the non-random sample selection issue (with a non-trivial proportion of observations missing) would undermine the validity of the results. At least, some tests/checks should be performed to see how missing data impacts your results.

Reviewer #4: (No Response)

7. PLOS authors have the option to publish the peer review history of their article (what does this mean?). If published, this will include your full peer review and any attached files.

Reviewer #2: **Yes: **haisong dong

Reviewer #3: **Yes: **Qihui Chen

Reviewer #4: No

---

## [Author Response · Author response to Decision Letter 1]

1 Mar 2023

Response to reviewer #2 comments

Dear Reviewer Haisong Dong,

First, we thank you for your affirmation of our last revision. Secondly, your worries and suggestions this time are also crucial. According to your suggestions, we have made the following revisions on this manuscript: 

Question 1: Among the 125 city commercial banks in China, the Bank of Guangzhou, the Bank of Tianjin, the Bank of Beijing, the Bank of Shanghai, the Bank of Qilu, the Bank of Nanjing, the Bank of Hangzhou, the Bank of Dalian, the Bank of Shengjing, the Bank of Changsha, the Bank of Chongqing, the Bank of Chengdu and the Bank of Xiamen, etc are all city commercial banks with branches outside the province. (For example, as of September 2011, Bank of Beijing has 9 branches in Shanghai, Tianjin, Hangzhou, Shenzhen, Xi 'an, Changsha, Nanjing, Jinan and Nanchang, covering 9 provinces in China.).Taking “GDP” as an example, according to the author's reply (since urban commercial banks have multiple branches in the province, provincial GDP is used instead of city-level GDP), the above banks must use cross-provincial GDP, even China's GDP. Similarly, “Finawa” (financial awareness of the government) cannot be a provincial index.

If the author must add macroeconomic variables, the reviewer provides several solutions. Method 1 is that the author takes the city where the headquarters of the city commercial bank is located as the reference and adopts city-level indicators. In this way, data must be found again from relevant databases such as China City Yearbook to recalculate "GDP" and "Finawa" (financial awareness of the government). And the text should have the corresponding explanation. Method 2 is to delete these inter-provincial city commercial banks and adopt provincial indicators according to the author's idea, but the sample size may be reduced or even insufficient. Please use your own discretion.

Of course, if the author has a more reasonable reason that the data need not be recollected or changed, the reviewer thinks that "minor revision" is OK, otherwise, it must be "major revision".

Response 1：

Trans-provincial operation is a fact, but it must be admitted that the bank's main business area is in the province where it is registered, as shown in the following table. The focus of bank operation is almost all in the province where it is registered. Only Bank of Shanghai and Bank of Tianjin account for a smaller proportion of local operations. Even the Bank of Beijing, Bank of Dalian and Bank of Ningbo, which have many trans-provincial institutions, have more than half of their local operations, almost 70%. Of course, this is the conclusion based on the 2020 data analysis. In the earlier years, the proportion of local operations was higher. Therefore, we believe that it is reasonable to build macro indicators based on provincial level. The use of provincial macroeconomic indicators is also a common method of existing research (Sun et al., 2013; Liu et al., 2022a; 2022b).

Bank The proportion of banks operating in local provinces in 2020 (calculated according to the regional distribution of total loans, total assets or operating income disclosed in the bank's annual report)

Bank of Beijing 68%

Bank of Shanghai 44.68%

Bank of Tianjin 22.2%

Bank of Chongqing 78.7%

Bank of Guangzhou 96.1%

Qilu Bank 94.04%

Bank of Nanjing 82.43%

Bank of Hangzhou 69.77%

Bank of Dalian 55.44%

Shengjing Bank 92.5%

Bank of Changsha 97.16%

Bank of Chengdu 93.33%

Xiamen Bank 100%

Bank of Ningbo 66.98%

Bank of Guiyang 94.33%

Bank of Dongguan 92.02%

Huishang Bank 78.15%

Bank of Jiangsu 82.54%

We removed the control variable (GDP) in the regression in order to prove that the selection of macroeconomic indicators is reasonable. The regression results are shown in the following table. It can be found that the regression results are still robust and reliable.

 (1) (2) (3) (4) (5) (6) (7) (8)

 Ln_Z Ln_Z ROA ROA CAR CAR σ(ROA) σ(ROA)

Priv 0.508*** 1.577*** 0.211*** 0.538*** -0.037*** -0.184*** -0.153** -0.222***

 (3.46) (6.01) (10.32) (2.84) (-3.82) (-7.81) (-2.54) (-3.79)

Finawa -0.007 0.225*** -0.001 -0.148***

 (-0.03) (2.99) (-0.17) (-4.44)

Priv×Finawa -1.700*** -0.550** 0.227*** 0.168*

 (-4.03) (-2.29) (7.19) (1.93)

Ln_Zt-1 0.879*** 0.808*** 

 (18.18) (20.13) 

ROAt-1 0.342*** 0.405*** 

 (13.13) (17.63) 

CARt-1 0.204*** 0.173*** 

 (10.44) (22.19) 

σ(ROA)t-1 0.659*** 0.402***

 (11.56) (24.58)

bank individual effect/year effect/X Yes Yes Yes Yes Yes Yes Yes Yes

N 466 466 658 658 681 681 480 480

Wald 582.42 2728.76 1327.14 1465.76 215.35 912.05 352.83 1721.09

 0.0000 0.0000 0.0000 0.0000 0.0000 0.0000 0.0000 0.0000

Ar(1)-p 0.0000 0.0000 0.0062 0.0076 0.0531 0.0492 0.0016 0.0002

Ar(2)-p 0.4615 0.4519 0.3576 0.1849 0.3738 0.5107 0.0513 0.3706

Sargan-p 0.7268 0.2196 0.3339 0.3856 0.4661 0.1156 0.3812 0.5513

Further, we removed the sample of CCBs (Bank of Shanghai, Bank of Tianjin, Bank of Beijing, Bank of Dalian and Bank of Ningbo) with a large proportion of operations in other provinces. The regression results are shown in the following table. It can be found that the regression results are still robust and reliable.

 (1) (2) (3) (4) (5) (6) (7) (8)

 Ln_Z Ln_Z ROA ROA CAR CAR σ(ROA) σ(ROA)

Priv 0.505*** 1.449*** 0.188*** 0.523*** -0.027*** -0.184*** -0.149** -0.244***

 (3.37) (6.33) (6.82) (2.90) (-2.71) (-8.30) (-2.45) (-5.45)

Finawa 0.115 0.257*** -0.007 -0.018

 (0.66) (3.40) (-1.59) (-0.45)

Priv×Finawa -1.560*** -0.521** 0.226*** 0.160*

 (-4.20) (-2.26) (7.72) (1.88)

Ln_Zt-1 0.889*** 0.793*** 

 (18.62) (24.73) 

ROAt-1 0.354*** 0.391*** 

 (12.70) (17.53) 

CARt-1 0.199*** 0.172*** 

 (9.73) (22.26) 

σ(ROA)t-1 0.649*** 0.581***

 (11.57) (64.10)

bank individual effect/year effect/X Yes Yes Yes Yes Yes Yes Yes Yes

N 442 442 625 625 648 648 456 456

Wald 692.08 3735.11 1139.48 1535.15 199.80 953.03 363.88 14594.75

 0.0000 0.0000 0.0000 0.0000 0.0000 0.0000 0.0000 0.0000

Ar(1)-p 0.0000 0.0000 0.0107 0.0085 0.0584 0.0488 0.0013 0.0004

Ar(2)-p 0.2148 0.2207 0.5133 0.2053 0.4779 0.5442 0.0537 0.2330

Sargan-p 0.8019 0.2558 0.4251 0.4626 0.3159 0.1653 0.2633 0.5322

Suggestion 1: According to the author's reply to Suggestion 1 of Reviewer 2, the government's financial consciousness (Finawa) does not affect the stable operation of banks (Ln_Z) through private capital holding (Priv), that is, it is not an intermediary effect, but a moderating effect. On the basis of analyzing the influence of "Priv" on "Ln_Z", the reviewer further studies whether "Finawa" changes the influence of "Priv", which is agreed by the reviewer.

However, the title "Financial awareness of government and steady operation of banks" of the author's research hypothesis 2 and “research hypothesis 2” do not seem to reflect the moderating effect of "Finawa". Instead, introduce the direct effect of "Finawa" on the steady operation of banks (ln-Z). Therefore, the reviewer thinks it is necessary to revise the title of research hypothesis 2, research Hypothesis 2, and even the content analysis. Please modify it.

Response 1：

We have revised the title of research hypothesis 2 and research Hypothesis 2.

Suggestion 2: According to Suggestion 3 of Reviewer 2, the author added "YEAR" to “the model expression” to represent the year effect.

But please revise the wording. Line 184: "Of course, we also control the trend effect of year, YEAR." should be changed to "YEARt is year effect"; Line 178: "Wi is the individual fixed effect of the banks" is changed to "Wi is banks individual effect" and delete the word "fixed". Please modify it.

Response 2：

We have revised the relevant words.

Suggestion 3: According to Suggestion 6 of Reviewer 2, the author adds a subsection "Stability test of Variable Data". But according to the author's modification, does “Fisher test” refer to “Fisher-ADF test” or “Fisher-PP test”? Please specify.

In Table 3, Column 1 should list the names of all variables, and the abbreviations of the variables before and after should be the same. For example, "P" should be changed to "Priv", "Z" should be changed to "Ln-Z", that is, to add logarithms, and so on. The reviewer even has the feeling that the author did not really test the Stability of the variables, which led to these errors. Please modify it.

Response 3：

Our "Fisher test" refers to "Fisher - ADF test". We did not specify the meaning of the symbols in the first column of Table 3 in detail in the manuscript, which is easy to be misunderstood. Here, P, Z, L and Pm are the four statistics of "Fisher - ADF test", namely Inverse Chi-squared P, Inverse normal Z, Inverse logit t L and Modified inverse chi-squared Pm, not variables. We have added relevant instructions to the manuscript.

Suggestion 4: According to Suggestion 4 of Reviewer 2, it is necessary to explain the application conditions of SYS-GMM method (such as autocorrelation test of random disturbance items and overrecognition test), but the reviewer found that the author did not explain in the paper (although there are relevant results in the table), please make a supplementary explanation.

The reviewer thinks this part is necessary for the following reasons: the empirical results of fixed effects of this paper show that the direct effect of "Priv" is not significant, and the moderating effect of "Finawa" is not significant, and the author wants to reach the research conclusion of this paper, so the SYS-GMM method is adopted. It must be explained here whether the application conditions of SYS-GMM method are met. In this way, the empirical results of SYS-GMM studied in this paper, “"Priv" has a direct effect and "Finawa" has a moderating effect”, are reasonable. Please modify it.

Response 4：

We have explained the application conditions of the SYS-GMM method based on the regression results in Table 4 and Table 5.

Suggestion 5: In "Table 4～Table 10", it is necessary to increase the "bank individual effect"; Time effect, should be "year effect", not "year"; The "control variables" should be changed to "X" (because "X" represents all control variables), and it is suggested that "X", "year effect" and "bank individual effect" should be displayed in three lines in "Table 4～Table 10", and the empirical results should be represented by "Yes". Please modify it.

Response 5：

We have revised the presentation of regression results.

Suggestion 6: In the "Future prospects" of this paper, please point out the limitations of this research, so as to lead to the future research direction of this topic. Please add clarification.

Response 6：

We have added the limitations of the paper.

Sun J, Harimaya K, Yamori N. Regional economic development, strategic investors, and efficiency of Chinese city commercial banks. Journal of Banking & Finance. 2013; 37(5): 1602-1611. https://doi.org/10.1016/j.jbankfin.2012.12.013

Liu J, Xu L, Zhang Q. The influence of the largest private shareholder on bank loans: Evidence from China. PLoS ONE. 2022a; 17(10): e0276877. https://doi.org/10.1371/journal.pone.0276877

Liu J, Zhang Q, Xiao C. Large private shareholders, industrial policies and industrial loans of city commercial banks: Evidence from China. PLoS ONE. 2022b; 17(12): e0278654. https://doi.org/10.1371/journal.pone.0278654

Response to reviewer #3 comments

Dear Qihui Chen,

We are encouraged by your approval of our revision. We express our profound gratitude for this. The problem of sample self-selection is the key to the reliability of empirical analysis, and we attach great importance to it. Our response to this problem is as follows:

In addition to the influence of the lag-term set by the model, the other reason is that the Z value is obtained from the moving average of the three years before and after the ROA and CAR, which results in the Z value of the first and last year of the sample being the missing value, so the observations of Ln_Z is less than that of ROA and CAR. The same is true of Volatility of ROA. There is no serious sample self-selection problem in this processing method.

However, in order to increase the reliability of the empirical results, we test whether there is a problem of sample self-selection according to the idea of Beckman (1979) two-stage estimation. The results are shown in the following table.

The regression results of the first stage are consistent with the first column of Table 8 in the manuscript. By calculating the Inverse Mill Ratio index (IMR) and adding it to the second stage of regression, the results show that the coefficients of IMR are not significant, indicating that there is no serious problem of sample self-selection.

 (1) (2) (3) (4) (5)

 Priv Ln_Z ROA CAR σ(ROA)

Priv 0.490*** 0.232*** -0.029*** -0.139**

 (3.33) (7.99) (-3.48) (-2.27)

Ln_Zt-1 0.875*** 

 (19.16) 

ROAt-1 0.380*** 

 (14.27) 

CARt-1 0.175*** 

 (14.75) 

σ(ROA)t-1 0.675***

 (14.69)

Priv_iv 0.499** 

 (2.22) 

Finawa 0.634*** 

 (5.87) 

IMR -1.271 0.134 -0.014 0.435

 (-1.34) (0.63) (-0.48) (1.47)

bank individual effect/year effect/X Yes Yes Yes Yes Yes

N 694 466 657 680 480

Wald 738.29 1187.63 393.41 606.25

 0.0000 0.0000 0.0000 0.0000

Ar(1)-p 0.0000 0.0073 0.0619 0.0009

Ar(2)-p 0.4269 0.4861 0.6445 0.3460

Sargan-p 0.5345 0.4014 0.1718 0.0667

Heckman J. Sample selection bias as a specification error. Econometrica. 1979; 47(1): 153-161. http://dx.doi.org/10.2307/1912352

---

## [Decision Letter · Decision Letter 2]

13 Mar 2023

PONE-D-22-34925R2Private capital holding, financial awareness of government and steadying operation of banks: Evidence from ChinaPLOS ONE

Dear Dr. Jie Liu,

Thank you for submitting your manuscript to PLOS ONE. After careful consideration, we feel that it has merit but does not fully meet PLOS ONE’s publication criteria as it currently stands. Therefore, we invite you to submit a revised version of the manuscript that addresses the points raised during the review process.

Please submit your revised manuscript by Apr 27 2023 11:59PM. If you will need more time than this to complete your revisions, please reply to this message or contact the journal office at plosone@plos.org. Please include the following items when submitting your revised manuscript:A rebuttal letter that responds to each point raised by the academic editor and reviewer(s). You should upload this letter as a separate file labeled 'Response to Reviewers'.A marked-up copy of your manuscript that highlights changes made to the original version. You should upload this as a separate file labeled 'Revised Manuscript with Track Changes'.An unmarked version of your revised paper without tracked changes. You should upload this as a separate file labeled 'Manuscript'.If applicable, we recommend that you deposit your laboratory protocols in protocols.io to enhance the reproducibility of your results. Protocols.io assigns your protocol its own identifier (DOI) so that it can be cited independently in the future. For instructions see: https://journals.plos.org/plosone/s/submission-guidelines#loc-laboratory-protocols. Additionally, PLOS ONE offers an option for publishing peer-reviewed Lab Protocol articles, which describe protocols hosted on protocols.io. Read more information on sharing protocols at https://plos.org/protocols?utm_medium=editorial-email&utm_source=authorletters&utm_campaign=protocols.

We look forward to receiving your revised manuscript.

Kind regards,

María del Carmen Valls Martínez, Ph.D.

Academic Editor

PLOS ONE

Journal Requirements:

Reviewers' comments:

Reviewer's Responses to Questions

**Comments to the Author**

1. If the authors have adequately addressed your comments raised in a previous round of review and you feel that this manuscript is now acceptable for publication, you may indicate that here to bypass the “Comments to the Author” section, enter your conflict of interest statement in the “Confidential to Editor” section, and submit your "Accept" recommendation.

Reviewer #2: (No Response)

Reviewer #3: All comments have been addressed

2. Is the manuscript technically sound, and do the data support the conclusions?

Reviewer #2: Yes

Reviewer #3: Yes

3. Has the statistical analysis been performed appropriately and rigorously? 

Reviewer #2: No

Reviewer #3: Yes

4. Have the authors made all data underlying the findings in their manuscript fully available?

Reviewer #2: Yes

Reviewer #3: Yes

5. Is the manuscript presented in an intelligible fashion and written in standard English?

Reviewer #2: Yes

Reviewer #3: Yes

6. Review Comments to the Author

Reviewer #2: Review comments: minor revision

Thank the authors for explaining and modifying of the reviewer's questions and suggestions, but some details or parts need to be improved, specifically as follows:

Suggestion 1: The title of hypothesis 2 is suggested to be revised to "Financial awareness of government, Private capital holding, and steady operation of banks". On the one hand, it can reflect the moderating effect of "Finawa"; on the other hand, it is consistent with the format of the title "Ownership structure and steady operation of banks" in hypothesis 1.

Or the title of hypothesis 2, "The moderating effect of the government's financial awareness", should be retained unchanged. The title of research hypothesis 1 is modified to "Direct Impact of Private capital holding", which may be more consistent in format. Please consider.

Suggestion 2: Line 183:"Of course, we also control the year effect, YEAR.", the author uses “YEAR” to control the time effect, this expression, “YEAR” should be the time trend term, the model expression is not "YEARt", but "λYEAR", that is, there should be a constant term λ before "YEAR"; Corresponding "Table 4-Table10", "year effect" should be modified to "YEAR".

If line 183 is modified to "YEARt is year effect", it means that "YEARt" is the time effect, the model expression is unchanged, and the corresponding "Table 4-Table10", "year effect" is also unchanged. Based on the empirical results, the authors confirm whether it is "YEAR" or "YEARt" and make corresponding modifications. Please modify it.

Suggestion 3: In the "Stability test of variable" section, it should be that all variables should undergo the stability test, that is, all variables should undergo the Fisher-ADF test.

Only when all variables are stationary or there is a co-integration relationship between all variables can "pseudo-regression" be avoided and the accuracy of empirical analysis be guaranteed. Authors should not just perform Fisher-ADF tests for "Ln Z". Please modify it.

Suggestion 4: One of the applicable conditions of SYS-GMM method is the autocorrelation test of random disturbance terms, which is tested by whether the difference of random disturbance terms exists first-order and second-order autocorrelation. The autocorrelation test of random disturbance item can only be passed if the difference of random disturbance item has first-order autocorrelation but not second-order autocorrelation. Therefore, the authors should add that the difference of random disturbance item has first-order autocorrelation in Line 229. Please add clarification.

Also, “This means that the random disturbance term has no autocorrelation and pass the over-identification test.”（Line 231）is some ambiguity in the translation. Please modify it.

Suggestion 5: In section "Results", Table 4 should be removed, as well as the analysis related to Table 4. The reason is that, firstly, Table 5 can be used as the benchmark regression result of this paper, which can be used to verify both hypothesis 2 and hypothesis 1. So table 4 is redundant. Secondly, the two stability tests carried out in the subsection "Robustness checks" are also comparing with Table 5 as the benchmark regression results, which also mean that Table 4 is redundant. In addition, validation analysis of research hypothesis 1 should be supplemented in Table 5. Please modify it.

Reviewer #3: After two rounds of revisions, the paper has been sufficiently improved. I have no further comments.

7. PLOS authors have the option to publish the peer review history of their article (what does this mean?). If published, this will include your full peer review and any attached files.

Reviewer #2: **Yes: **Haisong Dong

Reviewer #3: No

---

## [Author Response · Author response to Decision Letter 2]

3 Apr 2023

Response to reviewer #2 comments

Dear Reviewer Haisong Dong,

Once again, thank you for your affirmation of the revisions we have made. Your suggestions are really helpful for revising and improving our paper. According to your suggestions, we have made the following revisions on this manuscript: 

Suggestion 1: The title of hypothesis 2 is suggested to be revised to "Financial awareness of government, Private capital holding, and steady operation of banks". On the one hand, it can reflect the moderating effect of "Finawa"; on the other hand, it is consistent with the format of the title "Ownership structure and steady operation of banks" in hypothesis 1.

Or the title of hypothesis 2, "The moderating effect of the government's financial awareness", should be retained unchanged. The title of research hypothesis 1 is modified to "Direct Impact of Private capital holding", which may be more consistent in format. Please consider.

Response 1：

We have changed the title of hypothesis 2 to "Financial awareness of government, Private capital holding, and steady operation of banks".

Suggestion 2: Line 183:"Of course, we also control the year effect, YEAR.", the author uses “YEAR” to control the time effect, this expression, “YEAR” should be the time trend term, the model expression is not "YEARt", but "λYEAR", that is, there should be a constant term λ before "YEAR"; Corresponding "Table 4-Table10", "year effect" should be modified to "YEAR".

If line 183 is modified to "YEARt is year effect", it means that "YEARt" is the time effect, the model expression is unchanged, and the corresponding "Table 4-Table10", "year effect" is also unchanged. Based on the empirical results, the authors confirm whether it is "YEAR" or "YEARt" and make corresponding modifications. Please modify it.

Response 2：

The YEAR here is the year effect, which we labeled "yeart".

Suggestion 3: In the "Stability test of variable" section, it should be that all variables should undergo the stability test, that is, all variables should undergo the Fisher-ADF test.

Only when all variables are stationary or there is a co-integration relationship between all variables can "pseudo-regression" be avoided and the accuracy of empirical analysis be guaranteed. Authors should not just perform Fisher-ADF tests for "Ln Z". Please modify it.

Response 3：

We have added stability test for other variables.

Suggestion 4: One of the applicable conditions of SYS-GMM method is the autocorrelation test of random disturbance terms, which is tested by whether the difference of random disturbance terms exists first-order and second-order autocorrelation. The autocorrelation test of random disturbance item can only be passed if the difference of random disturbance item has first-order autocorrelation but not second-order autocorrelation. Therefore, the authors should add that the difference of random disturbance item has first-order autocorrelation in Line 229. Please add clarification.

Also, “This means that the random disturbance term has no autocorrelation and pass the over-identification test.”（Line 231）is some ambiguity in the translation. Please modify it.

Response 4：

As you pointed out, the prerequisite of SYS-GMM is that there is no autocorrelation in the differences of random disturbance terms. Of course, the most critical test is to verify whether there is second-order autocorrelation. Theoretically, even if the assumption is satisfied, there still is a first-order autocorrelation, but in fact, due to the factual situation of the data, the first-order autocorrelation may not exist.

Meanwhile, we replaced “This means that the random disturbance term has no autocorrelation and pass the over-identification test.” with “This means that the random disturbance term has no autocorrelation, and the model setting passes the over-identification test.”

Suggestion 5: In section "Results", Table 4 should be removed, as well as the analysis related to Table 4. The reason is that, firstly, Table 5 can be used as the benchmark regression result of this paper, which can be used to verify both hypothesis 2 and hypothesis 1. So table 4 is redundant. Secondly, the two stability tests carried out in the subsection "Robustness checks" are also comparing with Table 5 as the benchmark regression results, which also mean that Table 4 is redundant. In addition, validation analysis of research hypothesis 1 should be supplemented in Table 5. Please modify it.

Response 5：

Indeed, Table 5 is more important than Table 4, but we need Table 4 to directly confirm Hypothesis 1. In contrast, the role of Table 5 is to reflect regulatory effects. Therefore, we would like you to agree that we retain Table 4.

---

## [Decision Letter · Decision Letter 3]

10 Apr 2023

PONE-D-22-34925R3Private capital holding, financial awareness of government and steadying operation of banks: Evidence from ChinaPLOS ONE

Dear Dr. Liu,

Thank you for submitting your manuscript to PLOS ONE. After careful consideration, we feel that it has merit but does not fully meet PLOS ONE’s publication criteria as it currently stands. Therefore, we invite you to submit a revised version of the manuscript that addresses the points raised during the review process.

We look forward to receiving your revised manuscript.

Kind regards,

María del Carmen Valls Martínez, Ph.D.

Academic Editor

PLOS ONE

Journal Requirements:

Reviewers' comments:

Reviewer's Responses to Questions

**Comments to the Author**

1. If the authors have adequately addressed your comments raised in a previous round of review and you feel that this manuscript is now acceptable for publication, you may indicate that here to bypass the “Comments to the Author” section, enter your conflict of interest statement in the “Confidential to Editor” section, and submit your "Accept" recommendation.

Reviewer #2: (No Response)

2. Is the manuscript technically sound, and do the data support the conclusions?

Reviewer #2: Yes

3. Has the statistical analysis been performed appropriately and rigorously? 

Reviewer #2: N/A

4. Have the authors made all data underlying the findings in their manuscript fully available?

Reviewer #2: Yes

5. Is the manuscript presented in an intelligible fashion and written in standard English?

Reviewer #2: Yes

6. Review Comments to the Author

Reviewer #2: 1. In Line 184, please change "Of course, we also control the year effect, yeart." to "Of course, yeart is year effect."

2、In Line 232, please change “Arellano-Bond test for zero autocorrelation in first-differenced errors shows that all Ar(2)-p are greater than 0.05,” to “In the Arellano-Bond test, the difference of random disturbance terms shows that all Ar(1)-p is less than 0.1, and all Ar(2)-p is greater than 0.1, that is, the difference of random disturbance terms has first-order autocorrelation and no second-order autocorrelation;”.

3、In the section "Results", Table 4 should be removed, as well as related analysis in Table 4. In addition, verification analysis of research hypothesis 1 should be supplemented in Table 5. The reasons are as follows: First, Table 5 can be used as the benchmark regression result of this paper, which can be used for the verification of Hypothesis 2 and Hypothesis 1. Second, The two stability tests carried out in the section "Robustness checks" are also carried out with Table 5 as the benchmark regression results for comparative testing.

7. PLOS authors have the option to publish the peer review history of their article (what does this mean?). If published, this will include your full peer review and any attached files.

Reviewer #2: **Yes: **Haisong Dong

---

## [Author Response · Author response to Decision Letter 3]

24 Apr 2023

Response to reviewer #2 comments

Dear Reviewer Haisong Dong,

Thank you again for your suggestion. Your suggestions are really helpful for revising and improving our paper. According to your suggestions, we have made the following revisions on this manuscript:

Suggestion 1: In Line 184, please change "Of course, we also control the year effect, yeart." to "Of course, yeart is year effect."

Response 1：

We have made this change.

Suggestion 2: In Line 232, please change “Arellano-Bond test for zero autocorrelation in first-differenced errors shows that all Ar(2)-p are greater than 0.05,” to “In the Arellano-Bond test, the difference of random disturbance terms shows that all Ar(1)-p is less than 0.1, and all Ar(2)-p is greater than 0.1, that is, the difference of random disturbance terms has first-order autocorrelation and no second-order autocorrelation;”

Response 2：

We have made this change.

Suggestion 3: In the section "Results", Table 4 should be removed, as well as related analysis in Table 4. In addition, verification analysis of research hypothesis 1 should be supplemented in Table 5. The reasons are as follows: First, Table 5 can be used as the benchmark regression result of this paper, which can be used for the verification of Hypothesis 2 and Hypothesis 1. Second, The two stability tests carried out in the section "Robustness checks" are also carried out with Table 5 as the benchmark regression results for comparative testing.

Response 3：

We have made this change. The specific content is reflected in lines 206 to 246 of the revised manuscript.

---

## [Editor Report · Decision Letter 4]

27 Apr 2023

Private capital holding, financial awareness of government and steadying operation of banks: Evidence from China

PONE-D-22-34925R4

Dear Dr. Jie Liu,

We’re pleased to inform you that your manuscript has been judged scientifically suitable for publication and will be formally accepted for publication once it meets all outstanding technical requirements.

Kind regards,

María del Carmen Valls Martínez, Ph.D.

Academic Editor

PLOS ONE
---

## [Editor Report · Acceptance letter]

2 May 2023

PONE-D-22-34925R4 

Private capital holding, financial awareness of government and steadying operation of banks: Evidence from China 

Dear Dr. Liu:

I'm pleased to inform you that your manuscript has been deemed suitable for publication in PLOS ONE. Congratulations! Your manuscript is now with our production department. 

Kind regards, 

on behalf of

Dr. María del Carmen Valls Martínez 

Academic Editor

PLOS ONE